# Immunogenicity and Safety Profile of Two Adjuvanted-PD-L1-Based Vaccine Candidates in Mice, Rats, Rabbits, and Cynomolgus Monkeys

**DOI:** 10.3390/vaccines13030296

**Published:** 2025-03-11

**Authors:** Camila Canaán-Haden, Javier Sánchez-Ramírez, Rafael Martínez-Castillo, Mónica Bequet-Romero, Pedro Puente-Pérez, Isabel Gonzalez-Moya, Yunier Rodríguez-Álvarez, Marta Ayala-Ávila, Jorge Castro-Velazco, Olivia Cabanillas-Bernal, Marco A. De-León-Nava, Alexei F. Licea-Navarro, Yanelys Morera-Díaz

**Affiliations:** 1Center for Genetic Engineering and Biotechnology (CIGB), P.O. Box 6162, Playa Cubanacán, Havana 10600, Cuba; camila.canaanhaden@cigb.edu.cu (C.C.-H.); javier.sanchezramirez@dcu.ie (J.S.-R.); rafael.martinez@cigb.edu.cu (R.M.-C.); monica.bequet@cigb.edu.cu (M.B.-R.); pedro.puente@cigb.edu.cu (P.P.-P.); isabel.gonzalez@cigb.edu.cu (I.G.-M.); yunier.rodriguez@cigb.edu.cu (Y.R.-Á.); marta.ayala@cigb.edu.cu (M.A.-Á.); jorge.castro@cigb.edu.cu (J.C.-V.); 2CONAHCYT—Innovation and Development Promotion Direction, Centro de Investigación Científica y Educación Superior de Ensenada (CICESE), Ensenada 22860, Mexico; cabanillas@cicese.mx; 3Biomedical Innovation Department, Centro de Investigación Científica y Educación Superior de Ensenada (CICESE), Ensenada 22860, Mexico; madeleon@cicese.mx (M.A.D.-L.-N.); alicea@cicese.mx (A.F.L.-N.)

**Keywords:** PKPD-L1, cancer vaccine, immunogenicity, preclinical safety

## Abstract

Background: The therapeutic blockade of the PD1/PD-L1 axis with monoclonal antibodies has led to a breakthrough in cancer treatment, as it plays a key role in the immune evasion of tumors. Nevertheless, treating patients with cancer with vaccines that stimulate a targeted immune response is another attractive approach for which few side effects have been observed in combination immunotherapy clinical trials. In this sense, our group has recently developed a therapeutic cancer vaccine candidate called PKPD-L1^Vac^ which contains as an antigen the extracellular domain of human PD-L1 fused to a 47 amino-terminal, part of the *LpdA* gene of *N. meningitides*, which is produced in *E. coli*. The investigation of potential toxicities associated with PD-L1 blockade by a new therapy in preclinical studies is critical to optimizing the efficacy and safety of that new therapy. Methods: Here, we describe immunogenicity and preliminary safety studies in mice, rats, rabbits, and non-human primates that make use of a 200 μg dose of PKPD-L1 in combination with VSSPs or alum phosphate to contribute to the assessment of potential adverse events that are relevant to the future clinical development program of this novel candidate. Results: The administration of PKPD-L1^Vac^ to the four species at the doses studied was immunogenic and did not result in behavioral, clinical, hematological, or serum biochemical changes. Conclusions: Therefore, PKPD-L1^Vac^ could be considered suitable for further complex toxicological studies and the way for its clinical evaluation in humans has been opened.

## 1. Introduction

The therapeutic blockade of the PD1/PD-L1 axis with monoclonal antibodies has led to one of the greatest breakthroughs in cancer treatment, as it plays a key role in the immune evasion of tumors [1]. To date, PD-1/PD-L1 checkpoint blockade is part of the standard of care for numerous malignancies [2,3]. Nevertheless, therapy with these successful monoclonal antibodies has been associated with several immune-related adverse events (irAEs) due to an excessive immune response. These irAEs range from mild to severe in various tissues, with the most common events affecting the skin, liver, lung, and gastrointestinal tract [4].

Treating cancer patients with vaccines that stimulate a targeted immune response is another attractive approach, with very few side effects having been observed in combination immunotherapy trials to date [5]. There are several active immunotherapy approaches to investigating PD1/PD-L1 as a promising alternative for cancer therapy [6,7,8,9]. In this sense, our group has recently developed a therapeutic cancer vaccine candidate called PKPD-L1^Vac^, which contains as an antigen the extracellular domain of human PD-L1 fused to a 47 amino-terminal, part of the *LpdA* gene of *N. meningitides*, which is produced in *E. coli.* Previous non-clinical studies have demonstrated that the immunization of mice with PKPD-1^Vac^ promoted not only a highly specific immune response but also a potent anti-tumoral effect in both syngeneic BALB/c and C57BL/6 mouse models [10].

Since self-antigens are weakly immunogenic, the PD-L1 vaccine requires the presence of adjuvants to elicit or enhance an adequate immune response. Due to their safety profile, aluminum adjuvants are often considered the “gold standard” against which newly developed adjuvants should be compared. Vaccines with aluminum-based adjuvants are widely used in immunization programs worldwide [11]. For this reason, the PKPD-L1 antigen, in the previously mentioned studies, was combined with aluminum phosphate (AP) in mice and non-human primates. However, as knowledge of immunology and the mechanisms of action of vaccine adjuvants has developed, the number of vaccines containing novel adjuvants has increased. We decided to also use very small synthetic proteoliposomes (VSSPs) as an adjuvant to enhance the specific antigen response and suppress the immunosuppression caused by the tumors [12,13]. Considering the aforementioned elements, in the present work, preclinical studies including immunogenicity and preliminary safety assessments were performed with both adjuvants.

In addition, the selection of appropriate animal models for safety testing improves the value of the results of animal-model studies. With this in mind, PKPD-L1^Vac^ has been studied in BALB/c and C57BL/6 mice, Wistar rats, New Zealand white rabbits, and *Chlorocebus aethiops sabaeus* monkeys. These experimental models progressively show a higher homology between their own and human PD-L1 extracellular domains: 73.52% in mice (UniProt ID: Q9EP73), 74.43% in rats (UniProt ID: D4AE25), 82.19% in rabbits (UniProt ID: G1SUI3), and 95.91% in monkeys (UniProt ID: A0A0D9R951). Here, we describe immunogenicity and preliminary safety studies in the aforementioned species with a 200 μg dose of PKPD-L1 in combination with aluminum phosphate or VSSPs to contribute to the assessment of potential adverse events relevant to the future clinical development program of this novel vaccine candidate.

## 2. Materials and Methods

### 2.1. Ethical Approval

Multiple animal models have been used to evaluate the immunogenicity and safety profiles of PKPD-L1^Vac^, including rodent models and non-human primate models. The studies in BALB/c and C57BL/6 mice, Wistar rats, and New Zealand rabbits were conducted under the approval of the CIGB Animal Care and Use Committee (Havana, Cuba) (CICUAL/CIGB/18010, CICUAL/CIGB/18009, 4SIE1702010, 6SIE1702003). Animals were adapted to laboratory conditions for at least 2 weeks, and fed with standard laboratory food, according to the species. African green monkeys were cared for according to the guidelines of the American Association for the Accreditation of Laboratory Animal Care (AAALAC) (CICUAL/CIGB/21050). Following the 3R principle, the number of monkeys was minimized in the experiments.

### 2.2. Animals

Female C57BL/6 and BALB/c mice weighing 17–19 g (6–8 weeks of age) were maintained at 5 animals per cage. Female Wistar rats weighing 250–270 g (9 weeks of age) were maintained at one animal per cage in contained areas. Female New Zealand rabbits weighing 1.5–2 kg (7–8 weeks of age) and healthy young African green monkeys (*Chlorocebus aethiops sabaeus*) weighing 3–7 kg (4–8 years old) were caged individually in specially tasked areas. All cages were labeled with the following information: principal investigator name, Animal Care and Use Protocol Number, species/strain/sex, date of birth, age and arrival, number of animals in cage, and group number.

All animals were purchased from the National Centre for Animal Breeding (CENPALAB, Havana, Cuba) and maintained in the animal facility of the Center for Genetic Engineering and Biotechnology (CIGB, Havana, Cuba) in accordance with the Cuban guidelines for the care and use of laboratory animals.

### 2.3. Vaccine Antigen and Adjuvants

The design, cloning, bacterial expression, and purification of the recombinant fusion protein PKPD-L1 were previously described [10]. Briefly, the extracellular domain of human PD-L1 was cloned and expressed in *E. coli* as an N-terminus fusion protein with the first 47 aminoacids of the *LpdA* gene from *N. meningitidis* using the pM238 plasmid. PKPD-L1 was purified using ion metal affinity chromatography (IMAC)/gel filtration. The purified protein was under the endotoxin limits for preclinical animal studies.

The adjuvant VSSPs were supplied by the Center of Molecular Immunology (CIM, Havana, Cuba). The aluminum phosphate (Brenntag Biosector, Frederikssund, Denmark) was formulated in Tris/HCL (10 mM, pH 7.4) and provided in sterile vials at 6.33 or 12.66 mg/mL by the Technological Development Department of the Center for Genetic Engineering and Biotechnology (TD-CIGB, Havana, Cuba).

### 2.4. Immunization Schedules

Mice were divided into four groups of seven animals each and received the following: (a) control animals received vehicle combined with VSSPs adjuvant preparations, (b) eight subcutaneous injections of 200 µg of the recombinant protein PKPD-L1 mixed with 100 µg of VSSPs on a weekly schedule, (c) control animals received vehicle combined with aluminum phosphate adjuvant preparations, and (d) four subcutaneous injections of 200 µg of the recombinant protein PKPD-L1 mixed with aluminum phosphate on a bi-weekly schedule (0.7 mg Al^3+^).

Rats and rabbits were randomly divided into three groups of between four and five animals each and received: (a) eight subcutaneous injections of 200 µg of the recombinant protein PKPD-L1 mixed with 200 µg of VSSPs on a weekly schedule, (b) four subcutaneous injections of 200 µg of the recombinant protein PKPD-L1 mixed with aluminum phosphate on a bi-weekly schedule (0.7 mg Al^3+^), and (c) control animals received vehicle preparations in combination with VSSPs and aluminum phosphate adjuvants.

Monkeys received five total subcutaneous injections of 200 µg of recombinant PKPD-L1 protein mixed with aluminum phosphate (0.7 mg Al^3+^ per dose). The first four immunizations consisted of bi-weekly vaccinations. The fifth immunization was administered 29 days after the fourth immunization (day 71) as a booster dose.

For all species, random numbers were generated using the Aleator Software “2N” computer tool developed at the University of Arkansas, USA.

### 2.5. ELISA Reagents

Skim milk powder (Cat# 1.15363.0500), bovine serum albumin (BSA) (Cat# K32491618-406) and TMB (Cat# 613544) were supplied by Merck (Darmstadt, Germany). Phosphate-buffered saline (PBS) (Cat# D1408) was provided by Sigma. Tween 20 solution (Cat# A1389) was supplied by AppliChem (Darmstadt, Germany).

The recombinant proteins human PD-L1/Fc chimera (R&D Systems, 156-B7, Nordenstadt, Germany), human PD-1/Fc chimera (R&D Systems, 1086-PD), and human CD80/Fc chimera (R&D Systems, 140-B1) were used as competitive ELISAs and human PD-L1 His tag (R&D Systems, 9049-B7) in indirect ELISAs described below. Streptavidin–peroxidase conjugate (Sigma, S5512, Marlborough, MA, USA) was used at 1/30,000 dilution.

Antibody anti-human PD-L1 Biotinylated was provided by R&D Systems (Cat# BAF156). Sheep anti-mouse IgG conjugated to HRP (Cat# A6782), HRP-conjugated goat anti-rat IgG polyclonal (Cat# A9037), HRP-conjugated goat anti-rabbit IgG polyclonal (Cat# A0545), goat anti-mouse IgG1 (Cat# M5532), goat anti-mouse IgG2a (Cat# M5657), goat anti-mouse IgG2b (Cat# M5782), goat anti-mouse IgG3 (Cat# M5907), goat anti-mouse IgM (Cat# M6157), and goat anti-mouse IgA (Cat# M6032) were supplied by Sigma. Commercially available monoclonal antibody specific to human PD-L1 (Biolegend, 329702, San Diego, CA, USA) was used as an inhibition positive control in competitive ELISAs. HRP-conjugated goat anti-human IgG polyclonal antibody (Jackson Immunoresearch Laboratories, 109-035-098, West Grove, PA, USA) was used at 80 ng/mL for detecting human serum IgG.

### 2.6. ELISA Procedures

Detailed ELISA protocols are described in Appendix A.

#### 2.6.1. ELISA for Detecting PD-L1 Specific IgG Human Antibodies

Plates (Costar, 3590, New York, NY, USA) were coated overnight at 4 °C with 1 µg/well of hPD-L1-His diluted in PBS (pH 7.4). Diluted sera from the experimental immunization groups were added to wells and were detected with HRP-conjugated sheep anti-mouse IgG antibody (for mice), HRP-conjugated goat anti-rat IgG polyclonal antibody (for rats), HRP-conjugated goat anti-rabbit IgG polyclonal antibody (for rabbits), or HRP-conjugated goat anti-human IgG polyclonal antibody (for monkeys). Rat 1.3 serum sample from group immunized with PKPD-L1 combined with VSSPs was rejected as “unsuitable for analysis” due to an insufficient volume for the ELISA procedures.

#### 2.6.2. Competitive ELISA for Measuring Serum-Mediated Inhibition of PD-1/PD-L1 Interaction

Competitive ELISA has been previously described in detail by Morera et al. [10]. Briefly, plates (Costar, 3590) were coated overnight at 4 °C with 10 ng/well of human PD-1/Fc chimera diluted in PBS (pH 7.4). Equal volumes of sera from immunized mice at a 1/25 dilution were mixed with 50 ng/well of human PD-L1/Fc chimera to allow the formation of immune complexes. After immune complex formation, free PD-L1 can bind to coated PD-1 and the binding PD-1/PD-L1 was detected with a biotinylated anti-human PD-L1 antibody followed by a streptavidin–peroxidase conjugate. The maximum binding of PD-L1 was recorded in the absence of any animal serum and a commercially available anti-human PD-L1 monoclonal antibody was used as an inhibition positive control. A similar procedure was conducted to evaluate the sera effects on CD80/PD-L1 interaction.

### 2.7. Specific Anti-PDL1 Antibody Class and Subclass

The specific anti-PDL1 antibody class and subclass were determined by ELISA using mouse antibody isotyping reagent (ISO2-1KT, Sigma) according to the manufacturer’s instructions. Briefly, ELISA plates (Costar, 3590) were coated with 1 µg/well of hPD-L1/Fc chimera in PBS (pH 7.4). Mouse sera were added to wells at a 1:100 dilution. The absorbance was read at 450 nm in a BioRad microtiter plate reader.

### 2.8. Follow-Up of Clinical, Behavioral and Hematological and Serum Biochemical Parameters

#### 2.8.1. Mice

All animals were observed daily for general health and well-being during the study period. Quantitative and qualitative parameters were documented at least every two days. The survival rate (%) was counted among the quantitative measurements, while the qualitative parameters included the changes in breathing and animal behavior, as well as the appearance of the feces and hair. The qualitative evaluation was performed using numerically defined categories (Appendix A). A week after the last PKPD-L1 antigen administration, the animals were euthanized and their main organs (heart, kidneys, lungs, and liver) were processed for macroscopic observation.

#### 2.8.2. Rats

Each animal was observed daily for general clinical signs. Body weight and a general clinical examination were performed before each vaccine administration until the fifth immunization with VSSPs and the second immunization with aluminum phosphate (day 28 of the study). The administration site was closely examined, and the animals were observed for the following signs: flaccidity, piloerection, prostration, involuntary movements, head shaking, ataxia, salivation, respiratory distress, lacrimation, hyperactivity or lethargy, in coordination, diarrhea, and any other sign. Seven days after the last immunization (fourth immunization with the combination of the antigen with aluminum phosphate and eighth immunization with the combination with VSSPs), the animals were euthanized and the organs were dissected and fixed in 10% formalin for histological evaluation.

#### 2.8.3. Rabbits

Each animal was observed daily for general clinical signs. Body weight, rectal temperature, and a general clinical examination were determined or performed prior to each vaccine administration. During the physical examination, particular emphasis was placed on detecting hepatomegaly, splenomegaly, or regional lymphadenopathy and abnormalities at the injection site. Blood samples were taken before the first vaccine administration and seven days after the last vaccination to determine hematological and serum biochemical parameters (Appendix A). The animals were euthanized one week after the last immunization and the organs were dissected and fixed in 10% formalin for histological evaluation.

Blood samples were analyzed using a Nihon Kohden hematology analyzer (Celltac model MEK6450J; Nishiochiai, Shinjuku-ku, Tokyo, Japan). A differential leukocyte count (measured as a percentage of white blood cells) including lymphocytes percentage, neutrophils percentage, and monocytes percentage was performed by staining peripheral blood slides with Giemsa reagent. Cells were counted using an optical microscope with an immersion lens (VistaVision, MO 000004, Zeiss, Wetzlar, Germany). Serum biochemical parameters were analyzed using a Cobas Integra 400 PLUS automated analyzer (Roche Diagnostic Systems, Rotkreuz, Switzerland).

#### 2.8.4. Non-Human Primates

Given their genetic diversity and differences in consanguinity, it was decided to use the non-human primates’ values or parameters as controls before starting treatment. During the study, the monkeys were observed twice daily for viability/mortality and changes in behavior, response to treatment therapy, or health status. On immunization days, a complete clinical examination was performed, including examination of the skin, hair, mucous membranes (conjunctiva, nose, mouth, ear, genitalia, and rectum), lymphatic system, urogenital system, digestive system, respiratory system, cardiovascular system, and nervous system. These examinations were carried out by two experienced veterinarians. Temperature and body weight were also examined. Blood samples were taken before the first vaccine administration and seven days after the last vaccination to determine hematological and serum biochemical parameters (Appendix A).

### 2.9. Rats and Rabbits Anatomopathological Studies and Organ Weights

After euthanasia, from rabbits, the spleen, ovary, thymus, kidneys, liver, heart, and lungs were collected. The organs were individually weighed on analytical balance for relative weight analysis according to the formula: organ weight (g)/animal weight (g) × 100. In rats, the skin (site of injection) was collected. The organs were then fixed in 10% neutral-buffered formalin. After fixation, the tissues were embedded in paraffin and 5 μm sections were sliced and stained with hematoxylin and eosin (HE) for microscopic examination. The histopathological evaluation was based on a qualitative analysis and was carried out to identify whether any pathological abnormalities or lesions were potential treatment-related effects.

### 2.10. Statistical Analysis

Direct or transformed (Log10) data that passed the normality test (Kolmogorov–Smirnov normality test) and showed variance homogeneity (Bartlett’s test) were included. Statistical analyses between two groups were performed using Student’s *t*-test, whereas comparisons of multiple groups were analyzed by one-way ANOVA using GraphPad Prism 8 for Windows version (GraphPad Software, San Diego, CA, USA). Data that did not fulfill normality and/or variance homogeneity test, even after transformations, were analyzed using the nonparametric test. Statistical significance is indicated as follows: *, *p* < 0.05; **, *p* < 0.01; ***, *p* < 0.001; ****, *p* < 0.0001; ns = not significant. *p* values < 0.05 were considered significant.

## 3. Results

### 3.1. Active Immunization with PKPD-L1^Vac^ Induces Specific Anti-hPD-L1 IgG Antibodies in Mice

To investigate the effects of the PD-L1-specific antibody response of the antigen’s combination with VSSPs or aluminum phosphate, we performed immunogenicity studies in four species. Firstly, PKPD-L1^Vac^ immunization experiments were done in mice with different MHC Class II backgrounds: BALB/c (H2-d) and C57BL/6 (H2-b). The animals were immunized with 200 μg of the antigen or the vehicle in combination with 100 μg of VSSPs or 0.7 mg of aluminum phosphate on weekly or bi-weekly schedules, respectively (Figure 1A). As shown in Figure 1B, induced IgG antibodies against human PD-L1 were observed in both mouse strains immunized with PKPD-L1 and VSSPs or alum phosphate adjuvants, with the antibody titers having a media value for BALB/c of 1: 441859 and for C57BL/6 of 1: 2725880; these were measured a week after the last immunization dose. In BALB/c, the antibody titers that were achieved did not show significant differences between the group immunized with the vaccine antigen combined with VSSPs and the group immunized with the antigen in combination with aluminum phosphate (*p* = 0.6818; unpaired *t*-test). In the C57BL/6 mouse strain, the anti-PD-L1 antibody titers from the animals immunized with the antigen formulated with alum phosphate were significantly higher than those found in the group immunized with the antigen with VSSPs (*p* = 0.0232; unpaired *t*-test).

Competitive ELISA assays were also performed to test the ability of the antibodies induced by PKPD-L1^Vac^ to hamper the interaction between PD-L1 and its cognate receptors. In this case, we evaluated three pools of sera from the seven animals (1:3; 2:2) whose antibody titers were evaluated. The PKPD-L1 immune sera efficiently blocked PD-L1/PD-1 and PD-L1/CD80 interactions in both mouse strains as compared to the sera from negative control animals (Figure 1C,D). In the C57BL/6 (*p* = 0.3816; unpaired *t*-test) and BALB/c (*p* > 0.9999; unpaired *t*-test) mouse strains, no differences in PD-1 neutralizing antibody activity were observed among the combination of the antigen with VSSPs or alum phosphate. In both mouse strains, anti-antibodies capable of neutralizing the binding of PD-L1/CD80 were also detected after immunization with PKPD-L1^Vac^. However, in the C57BL/6 immunized mice, a higher neutralizing activity was detected for the combination of the antigen with aluminum phosphate as compared to the formulation with VSSPs (*p* = 0.0114; unpaired *t*-test).

In order to further characterize the humoral immune response induced by immunization with PKPD-L1^Vac^, an indirect ELISA was performed using secondary goat anti-mouse antibodies that were specific for each Ig class and IgG subclass. As shown in Figure 1E–H, immunization with PKPD-L1 in VSSP or alum phosphate adjuvant induced PD-L1-specific antibodies of the IgG, IgM, and IgA classes, with IgG being the predominant immunoglobulin. Sera from the BALB/c mice immunized with PKPD-L1 in combination with the VSSP adjuvant predominantly induced anti-PD-L1 antibodies that corresponded to the TH2 IgG1 and TH1 IgG2b subclasses (Figure 1E) (IgG1 = IgG2b > IgG2a > IgG3). In this mouse strain, PKPD-L1 formulation in alum phosphate adjuvant predominantly induced antibodies of the IgG1 subclass, but IgG2b, IgG2a, and IgG3 were also detected (Figure 1F) (IgG1 > IgG2a = IgG2b = IgG3). On the other hand, sera from the C57BL/6 animals immunized with PKPD-L1 in the VSSPs or alum phosphate adjuvant induced PD-L1-specific antibodies of the TH2 IgG1 and TH1 IgG2b subclasses, although the rest of the evaluated subclasses were also detected (Figure 1G,H) (IgG1 = IgG2b > IgG2a = IgG3). The (IgG2a + IgG2b)/IgG1 ratio was determined as a surrogate of the Th1/Th2 response pattern. In the BALB/c mice immunized with PKPD-L1 combined with VSSPs, the ratio was 1.49, and for PKPD-L1 combined with aluminum phosphate, the ratio was 0.59. The ratio obtained with PKPD-L1 combined VSSPs was significantly higher than that obtained with the aluminum phosphate-adjuvanted formulation (*p* = 0.0043; unpaired *t*-test).

### 3.2. Active Immunization with PKPD-L1^Vac^ Induces Specific Anti-hPD-L1 IgG Antibodies in Rats and Rabbits

The immunogenicity of two PD-L1-based vaccine formulations was evaluated in Wistar rats and New Zealand white rabbits. The animals were immunized with 200 μg of the antigen or the vehicle in combination with 200 μg of VSSPs or 0.7 mg of aluminum phosphate on weekly or bi-weekly schedules, respectively (Figure 2A). As shown in Figure 2B, in both species immunized with PKPD-L1 and VSSP or alum phosphate adjuvant, IgG antibodies against human PD-L1 were induced, and were measured a week after the last immunization dose (Day 56). In rats, the titers showed a media value for the VSSPs of 1: 123893 and of 1: 719442 for the alum phosphate–antigen combination, without significant differences between the groups (*p* = 0.3208; unpaired *t*-test). In rabbits, the titers showed a media value for the VSSPs of 1: 232698 and for the alum phosphate of 1: 247185 and did not show significant differences between the groups that were immunized (*p* = 0.6013; unpaired *t*-test).

Competitive ELISA assays were also performed to test the ability of the antibodies induced by PKPD-L1^Vac^ to interfere with the interaction between hPD-L1 and its cognate receptors. PKPD-L1 immune sera efficiently blocked PD-L1/PD-1 and PD-L1/CD80 interactions in both species as compared to the sera from negative control animals (Figure 2C,D). In rats, no significant differences were observed in the activity of neutralizing antibodies for PD-1 (*p* = 0.1374; unpaired *t*-test) and for CD80 (*p* = 0.0564; unpaired *t*-test) between the combination of the antigen with VSSPs and the combination with alum phosphate. In the rabbits, the results were similar to those of the rats, and did not show significant differences in the activity of PD-1 (*p* = 0.7040; unpaired *t*-test) and CD80 (*p* = 0.7694; unpaired *t*-test) between the combination of the antigen with VSSPs or that of alum phosphate.

### 3.3. Active Immunization with Recombinant PKPD-L1 Antigen Induces Specific Anti-hPD-L1 IgG Antibodies in Non-Human Primates

To test the immunogenicity of the two PD-L1-based vaccines in an animal model closely related to humans, monkeys were immunized with PKPD-L1 antigen combined with aluminum phosphate using a bi-weekly schedule (Figure 3A). All immunized monkeys elicited anti-PD-L1 IgG antibodies after the first dose of 1: 9702 (day 14) (*p* = 0.0379; paired *t*-test) and the average titers reached levels of 1: 339 325 after the fourth dose of the induction phase (day 56) (*p* = 0.0003; paired *t*-test) with respect to the pre-immunization titers (day 0). A decrease in antibody titers was observed in the samples collected 63 (1: 226 154) (*p* = 0.0004; paired *t*-test) and 71 (1: 142 772) (*p* = 0.0012; paired *t*-test) days after the last dose of the induction phase, with significant differences being seen with respect to time zero. The immunizations were reinitiated on day 71, and, thereafter, the average titers reached 1:240 028 on day 78 (*p* = 0.0006; paired t-test) and peaked at 1:393 630 on day 85 (*p* = 0.0109; paired *t*-test), slightly exceeding the highest value at the end of the induction phase (Figure 3B).

The ability of the monkey sera to block the interaction of PD-L1/PD-1 was evaluated using a similar inhibition ELISA system to that reported for rats and rabbits. Figure 3C shows the average inhibition values (three replicates of each sample) obtained by dilutions of the sera from individual monkeys after the fourth immunization. The sera from all vaccinated monkeys showed inhibition of the PD-1/PD-L1 interaction. There were statistical differences (*p* = 0.0057, paired *t*-test) in the neutralizing activity at day 56 compared to the pre-vaccination values (Figure 3C).

### 3.4. Clinical, Hematological and Biochemical Parameters

#### 3.4.1. Mice

The mice remained healthy throughout the immunization schedule. The survival rate was 100% in all groups. The qualitative parameters evaluated in this study did not show any changes during the observational periods (Appendix A).

#### 3.4.2. Rats

All animals appeared generally healthy during the vaccination period. No changes in overall behavior, feeding, neuromuscular performance, or the appearance of fur in the immunized animals were reported. No significant differences were observed in terms of the body weight of the animals treated with the PKPD-L1 vaccine antigen compared to the vehicle control group plus adjuvants as well as between immunized groups (*p* > 0.05; two-way ANOVA and Sidak’s post-test (Figure 4).

#### 3.4.3. Rabbits

The rabbits remained healthy throughout the immunization process. No significant differences were observed in terms of body weight (Figure 5) and temperature (Appendix A) in the animals immunized with the PKPD-L1 vaccine antigen compared to the vehicle control group plus adjuvants. The temperature was measured every hour for a period of three hours after each immunization with PKPD-L1^Vac^. The temperature values were found to be within the physiological range, without differences among the experimental groups.

Hematological and biochemical parameters were analyzed in individual rabbits from each group and then the values from day 0 and day 56 were compared. Seven days after the final immunizations with PKPD-L1^vac^ with different adjuvants, no differences were observed as compared with the animals from the negative control group (*p* > 0.05; Wilcoxon test) (Table 1), and the parameters were found to be within normal ranges.

#### 3.4.4. Non-Human Primates (NHPs)

The animals remained healthy throughout the immunization schedule. No significant differences were observed in terms of body weight (Figure 6) and temperature (Appendix A) on the days evaluated (days 14, 28, 42, 56, 70, 77, 84) compared to day zero.

In addition, hematological and serum biochemical parameters were investigated in African green monkeys. The values before the first vaccine administration were compared with the values on the last day of the study (day 85). No effects on the hematologic values were observed (Table 2). Statistical analysis of the biochemical parameters between the first and last evaluations showed no significant differences. The values were above the pre-immunization limits. Nevertheless, all these parameters were reset to their original values on day 85 in all groups of the study.

### 3.5. Anatomopathological Studies in Rats and Rabbits

To detect possible pathologies associated with immunization with the PKPD-L1 vaccine antigen, anatomopathological studies were carried out in the Wistar rats and New Zealand rabbits.

#### 3.5.1. Rats

Macroscopic studies of systems did not reveal pathological modifications that were attributable to the vaccine (Figure 7). Histological changes at the inoculation site, as well as delimited small caseous nodules with a size of approximately 1 × 0.5 cm, were detected in all the animals that received alum as an adjuvant (vaccine-treated and negative controls). The nodules were located in the subdermis without affecting the dermis or epidermis (Figure 7B). These nodules were verified microscopically as aluminum granulomas, commonly observed when aluminum is used as an adjuvant (Figure 7A).

#### 3.5.2. Rabbits

Organ studies did not reveal pathological modifications attributable to the vaccine (Figure 8). The result of the evaluation of the histopathological diagnosis of the selected organs of the study animals (spleen, ovaries, thymus, kidneys, liver, heart, and lungs) did not show significant differences between the weight of the organs of the animals immunized with PKPD-L1 plus an adjuvant and the organs of those given the vehicle negative control. Nevertheless, significant differences (*p* = 0.0144, Tukey test) (higher values) were found in the relative liver weight in the rabbits immunized with 200 μg of PKPD-L1 in combination with VSSPs as compared with the values obtained for those from the negative control group (*p* < 0.05; one-way ANOVA and a Tukey’s post-test) (Figure 8B). Histological changes in the liver of three rabbits from this group were detected, revealing mild venous congestion and the presence of activated Kupffer cells. However, similar findings were present in control animals that received adjuvants alone.

## 4. Discussion

In recent years, the widespread use of immunotherapy has revolutionized the treatment of various types of cancer. In particular, immune checkpoint inhibitors targeting PD-L1 have shown promising results [2,3]. However, while the immune system is manipulated to trigger an antitumor response, an autoimmune or inflammatory response can also occur in normal tissue, which differs from the standard toxicities of chemotherapy due to its immune-based origin. A broad spectrum of immune-related adverse events could be one of the disadvantages of this type of oncotherapy [14].

Among the most frequently reported adverse events described for passive anti-PD-L1 antibodies are endocrine disorders (thyroid diseases such as hypothyroidism and hyperthyroidism, followed by pituitary and adrenal dysfunction), gastrointestinal symptoms (diarrhea, colitis, and nausea), pulmonary symptoms (pneumonia), skin symptoms (rash, pruritus, and leukoplakia), and musculoskeletal symptoms (arthralgia and myalgia). In the liver, elevated levels of aspartate aminotransferase (AST) or alanine aminotransferase (ALT) have been detected in some patients, indicating some type of liver damage [4].

Considering that passive immunotherapy strategies promote different scenarios than active immunotherapy, there is a possibility that similar risks will not occur with immunization [15]. Preliminary studies in mice showed that the anti-tumoral activity induced by PKPD-L1^Vac^ in combination with aluminum phosphate was mediated by humoral and cell-mediated responses [10]. However, the comparative performance of this immunogen with different adjuvants in pre-clinical animal models, the necessity of refinements to the vaccine timing, and the potential safety profile for the combinatorial use of this candidate were unclear. Therefore, vaccination with PKPD-L1^Vac^ was planned using two adjuvants of different chemical compositions that are either approved for clinical application or have been tested preclinically with promising results. The first of these adjuvants was aluminum phosphate, known to mainly promote Th2-type humoral responses to human protein subunit vaccines [16]. The second clinically tested adjuvant used was VSSPs, which preferentially promote Th1 cellular responses alongside the activation of CTL responses to peptides and proteins but can also stimulate the humoral response to different antigens [17]. The present study comprises a comparative study of the immunogenicity and safety of the PKPD-L1^Vac^ vaccine in combination with two adjuvants in mice, rats, rabbits and *Cynomolgus* Monkeys.

The results shown here indicate that the vaccine can elicit an IgG immune response capable of neutralizing the binding of PD-1 and CD80 to human PD-L1 in several species. Weekly vaccination regimens with the PKPD-L1 antigen using VSSPs as an adjuvant were generally comparable to the bi-weekly treatment in combination with alum, both in terms of the IgG titers observed and the inhibition of the binding of PD-L1 to its natural receptors. A higher response for both parameters was observed exclusively in the C57BL/6 mouse strain when the antigen was combined with aluminum phosphate. Aluminum-based adjuvants are known to preferentially prime Th2-type immune responses, even though recent studies show that, depending on the vaccination route, they can enhance both Th1 as well as Th2 cellular responses [16]. Conversely, while studies with murine models have shown that VSSP is able to induce humoral and cellular responses for different tumor-cell and peptide–antigen vaccines, this adjuvant is more prone to activating an antigen-specific CTL response. The VSSP induced in vitro IL 12 secretion, DC maturation, and antigen cross-presentation [18,19]. In the case of the BALB/c mice, being a strain classically chosen to evaluate the humoral response to antigens by its polarization towards a Th2 response, it becomes a little more difficult to find differences, as those observed between the two adjuvants were subtle, as they can both induce a humoral response [20,21]. While the C57BL/6 mice had relatively mild systemic humoral immune responses, the cell-mediated immunity activity observed in this strain was relatively higher than in other mouse strains. The ratio of B lymphocytes to T lymphocytes circulating in the C57BL/6 mice is inversely proportional to that of other inbred strains and there may be more subtle differences in the immune response that can be tuned [22]. These factors may help to explain why differences in the antibody response were observed exclusively in this strain, favoring the combination of the antigen with aluminum phosphate.

The presence IgG, IgM, and IgA antibodies was detected in mice that were immunized with the PD-L1-based vaccine, and IgG was indicated to be predominant. In the context of PD-L1 vaccination, most monoclonal antibodies approved for cancer treatment belong to this class of immunoglobulin, indicating the importance of biological effects mediated by this class of immunoglobulin [23]. In mice and humans, IgG1 (as well as IgG4 in humans) is associated with a Th2 profile, and the other subclasses are mainly associated with a Th1 profile [24]. The experiments showed a mixed Th1/Th2 pattern, corresponding to the induction of both humoral and cellular responses by the vaccine. According to the genetic background of each strain and the immunological patterns mentioned above, some differences were found in the predominant subclass. In the BALB/c mice, the combination with aluminum phosphate showed predominance towards a Th2 pattern, but in the C57BL/6 strain, a balanced Th1/Th2 pattern was clearly observed. Specifically for PKPD-L1^VAc^, immunogenic properties were favored by the aggregation of the fusion polypeptide [10]. For example, it has been demonstrated that the aggregation or particulation of an antigen could increase its overall immunogenicity by enhancing B-cell receptor cross-linking, leading to higher levels of B-cell activation and the targeting of internalized antigen to lysosomes with subsequent enhanced antigen presentation to T-cells [25].

The combination of PKPD-L1 with VSSP adjuvant showed a Th1 pattern in both mouse strains. Particularly interesting are the results in the BALB/c mouse strain. The BALB/c strain is a prototypical Th2-type mouse strain. Nevertheless, when the (IgG2a + IgG2b)/IgG1 ratio was determined, as a surrogate of the Th1/Th2 response pattern, mice immunized with the formulations adjuvanted in VSSPs had significantly higher values than the ones receiving Alum, an indirect indication of TH1 differentiation. These results are consistent with those shown by other authors using VSSPs as an adjuvant. VSSPs induce the secretion of inflammatory cytokines, such as IL-12, which induce the Th1 polarization of helper T-cells. They also stimulate the functional activity of specific cytotoxic T-cells, a phenomenon that is facilitated by the cross-presentation of exogenous antigens and the independence of helper T-cells in primary cytotoxic T-cell expansion [26].

The ability of the PKPD-L1 vaccine to induce PD-L1-specific antibodies is quite important in the context of its antitumor potential. Immunization-induced antibodies can neutralize PD-L1 expressed on tumor cells, which may interfere with the functions of this molecule in rescuing tumor apoptosis. Antibodies specific to PD-L1 could also have an impact on blocking the immunosuppressive effects of PD-L1, which are related to the inhibition of the maturation of DCs and T lymphocyte activation, and to the induction of MDSC and Treg recruitment [27,28].

One aspect to consider is the fact that the antibodies generated by immunization with the PKPD-L1 vaccine candidate rapidly decline and require a new dose of vaccine to be restored or reach higher levels. The immunization regimen in the vaccinated NHP showed that the number of antibodies decreased after immunization was discontinued. This fact could affect the safety of the vaccine antigen, and it is possible to prevent an undesired persistent response of anti-PD-L1 antibodies by simply avoiding new immunization.

Investigating potential toxicities associated with PD-L1 blockade by a new therapy in preclinical studies is critical to optimize efficacy and safety. The experiments detailed herein were conducted to determine whether the administration of the PKPD-L1 antigen in combination with two adjuvants and/or the anti-PD-L1 immune response induced by the immunization process cause detectable clinical, biochemical, and histological side effects. In particular, the non-clinical evaluation of vaccine safety characteristics in monkeys offers several advantages for clinical implementation. This species is more similar to humans than rodents and allows the use of clinically relevant vaccine doses [29].

A group of vaccines considered safe and that are currently used in clinical trials is known to have side effects such as fever, general malaise, headache, or discomfort at the site of administration [30,31,32,33,34]. The variables of body weight and body temperature are evaluated in a variety of studies on vaccines as they are sensitive and general indicators of the toxicity of pharmacological agents. The effects on weight usually range from a decrease in weight gain compared to controls to a loss of body weight over time, depending on the toxicity of the product tested. On the other hand, the temperature of animals in preclinical toxicology studies can help to predict the reactogenicity of a vaccine in clinical trials. In addition, temperature is a parameter that can be associated with the route of administration and the pain scale in animals [35]. The non-negative effects on animal body weight and temperature throughout the test suggest that the administration of PKPD-L1^Vac^ does not induce any changes in health status or signs of toxicity in the tested animals.

Hematology and blood biochemistry parameters are another tool to determine the health status of animals [36]. Their interpretation is useful in this study as they indicate the occurrence of physiological, hepatic, or renal alterations. In this work, the hematologic and blood chemistry parameters showed no significant changes in all species studied compared to controls, indicating no systemic immunologic changes. Both parameters are sensitive to immunotoxicity and are used to assess the evolution of animal health over time.

Furthermore, the macroscopic analysis of most organs revealed no changes in relative weight, surface lesions, or measured blood chemistry parameters. This was also consistent with the histopathology of the vital parenchymal organs (brain, heart, lungs, kidneys), which showed no clinically relevant differences between the controls and the PKPD-L1^Vac^-vaccinated animals. Only an increase in relative liver weight was observed in the rabbits immunized with 200 μg of PKPD-L1 in combination with VSSPs. Histologic evaluation revealed mild venous congestion and the presence of activated Kupffer cells (KCs). Similar histological findings were present in control animals and there were no side effects on enzymatic indicators of liver damage.

The human liver is an organ with a diverse array of immunologic functions. Hepatocytes, although they are not immune cells, express innate immune receptors and can serve as antigen-presenting cells. Also, they are an important reservoir of macrophages, with 80–90% of the body’s total macrophages consisting of Kupffer cells that reside within the hepatic sinusoids. They can become potent activators of T-cells in the presence of other pathogen-associated molecules or inflammatory cytokines. Also, the liver is rich in innate lymphocytes (e.g., NK and NKT cells), which are relatively abundant in the liver compared to other tissues of the body [37,38]. VSSPs are a nanoparticle drug composed of outer membrane vesicles of gram-negative bacteria Neisseria meningitidis and the GM3 ganglioside. The VSSPs stimulated peritoneal inflammation characterized by increased granulocytes and monocytes, including inflammatory monocytic cells [12,39]. On the other hand, a therapy aimed at blocking PD-L1 may have a positive effect on the immune system [40]. Therefore, the combination of the antigen with this potent immunomodulatory adjuvant could explain the activation of different immune cells in the liver, which increased the weight of the organ.

Another histologic finding following the administration of PKPD-L1^Vac^ includes focal inflammatory responses at the site of administration in treated animals that have been associated with the aluminum hydroxide adjuvant contained in the formulation. This agent enhances the immune response against antigens, but also causes inflammatory reactions that indirectly enhance the immune response.

The present study shows that vaccination with PKPD-L1 in mice, rats, rabbits, and monkeys in combination with different adjuvants at the doses studied did not cause behavioral, clinical, hematological, or serum biochemical changes and, therefore, can be considered suitable for further complex toxicological studies. This vaccine predominantly induces IgG, but also IgM and IgA antibodies specific to human PD-L1. The elicited antibodies block the interaction between PD-L1 and its receptors PD-1 and CD80. Both strategies, using either VSSPs or aluminum phosphate as an adjuvant combined with the PKPD-L1 antigen, deserve further evaluations in future phase I clinical trials.

## Figures and Tables

**Figure 1 vaccines-13-00296-f001:**
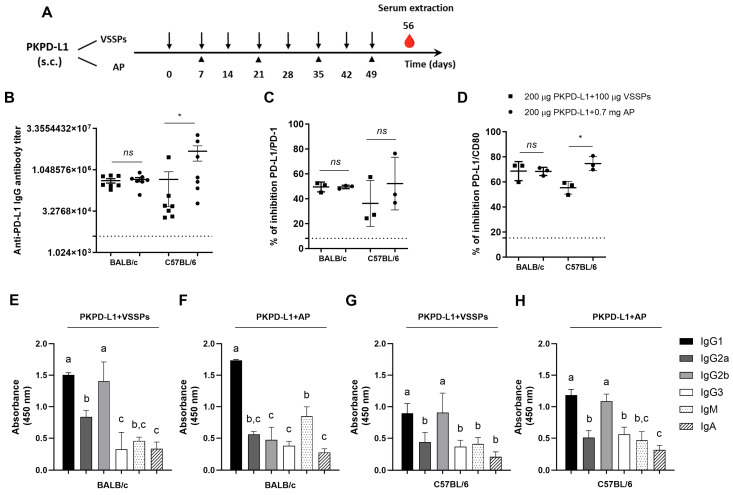
Specific humoral immune response in mice immunized with PKPD-L1. (**A**) Mice (*n* = 7) were immunized with the PKPD-L1 antigen in combination with VSSPs or aluminum phosphate (AP) for two months at weekly or two-week intervals, respectively. Control animals received only vehicle–adjuvant preparations and sera were extracted seven days after the last immunization for all groups. The black arrows indicate the immunization days for the weekly scheme and the head arrows the bi-weekly immunization. The blood drop represents the time point for sera extraction. (**B**) IgG antibody titer against human PD-L1, (**C**) inhibition of PD-1/PD-L1, and (**D**) CD-80/PD-L1 interactions by serum from vaccinated mice. (**E**,**F**) IgM- and IgA-class antibody response and response of specific IgG subclasses to human PD-L1 in BALB/c mice and (**G**,**H**) in C57BL/6 mice. The dashed line represents the average of the titer values or percentage inhibition of the negative control animals. In graph (**B**), each point represents the individual mice; data are presented as mean and the standard deviation of the mean (SEM). In graphs (**C**,**D**), each point represents percentage of inhibition calculated from duplicate samples of pools; data are presented as mean and the standard deviation (SD). *p*-values were calculated according to the unpaired *t*-test: ns *p* ≥ 0.05; * *p* < 0.05. In graphs (**E**–**H**), each column represents the mean absorbance at 450 nm calculated for serum samples from individual mice and the SD. Different letters indicate statistical significance according to one-way ANOVA and a Tukey’s post-test.

**Figure 2 vaccines-13-00296-f002:**
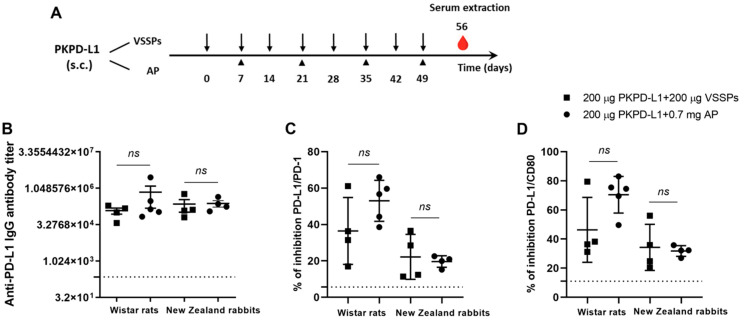
Specific humoral immune response in rats and rabbits immunized with PKPD-L1. (**A**) Rats (*n* = 5) and rabbits (*n* = 4) were immunized with the PKPD-L1 antigen in combination with VSSPs or aluminum phosphate (AP) for two months at weekly or two-week intervals, respectively. Control animals received only vehicle–adjuvant preparations and sera were extracted seven days after the last immunization for all groups. The black arrows indicate the immunization days for the weekly scheme and the head arrows the bi-weekly immunization days. The blood drop represents the time point for sera extraction. (**B**) IgG antibody titer against human PD-L1, (**C**) inhibition of PD-1/PD-L1 and (**D**) CD-80/PD-L1 interactions by serum from vaccinated rats and rabbits a week after the last immunization dose. The dashed line represents the average of the titer values or percentage inhibition of the negative control animals. In graph (**B**), each point represents the individual animals; data are presented as mean and the standard deviation of the mean (SEM). In graphs (**C**,**D**), each point represents the percentage of inhibition calculated from duplicate samples; data are presented as mean and the standard deviation (SD). *p*-values were calculated according to the unpaired *t*-test, ns *p* ≥ 0.05.

**Figure 3 vaccines-13-00296-f003:**
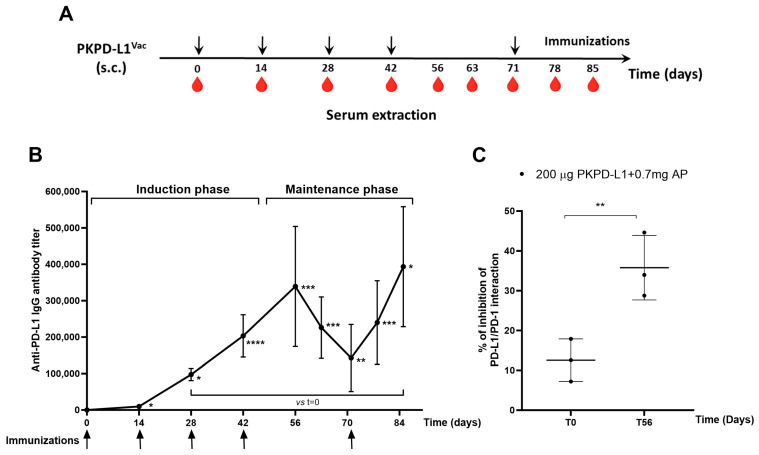
Specific humoral immune response in monkeys after PKPD-L1^Vac^ immunization. (**A**) Monkeys (*n* = 3) were immunized with the PKPD-L1 antigen in combination with aluminum phosphate for two months at two-week intervals and re-immunized 29 days later. Arrows indicate immunization days and the drop of blood represents the extraction of the sera before each immunization. (**B**) IgG antibody titer kinetics against human PD-L1. Each point represents the average titer calculated from duplicate samples of individual monkeys and the standard deviation of the mean titers (SEM). Arrows indicate immunization days. (**C**) Inhibition of PD-L1/PD-1 interaction by sera from non-human primates. Each point represents percentage of inhibition calculated from duplicate samples; data are presented as mean and the standard deviation (SD). *p*-values were calculated according to paired *t*-test, * *p* < 0.05; ** *p* < 0.01; *** *p* < 0.001; **** *p* < 0.0001.

**Figure 4 vaccines-13-00296-f004:**
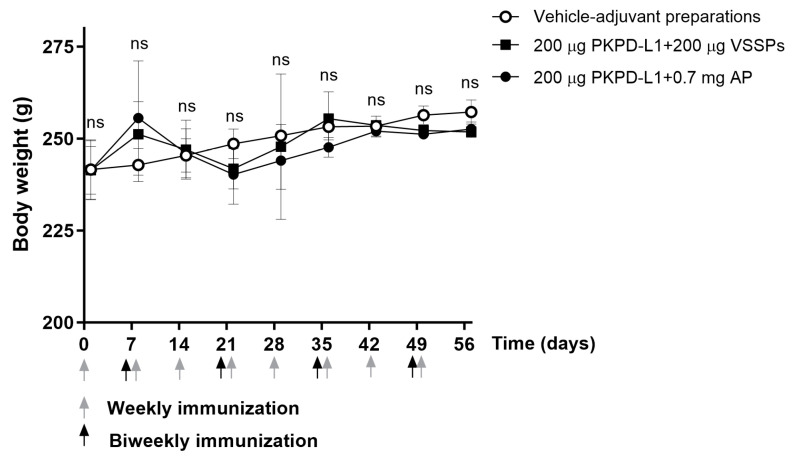
Time course of body weight in rats immunized with PKPD-L1. Each symbol represents the average and standard deviation of the body weight of the rats (*n* = 5) according to the day of the immunization schedule. Arrows indicate immunization days. No differences were detected between each immunization group (ns *p* ≥ 0.05, Two-way ANOVA, Sidak’s post-test).

**Figure 5 vaccines-13-00296-f005:**
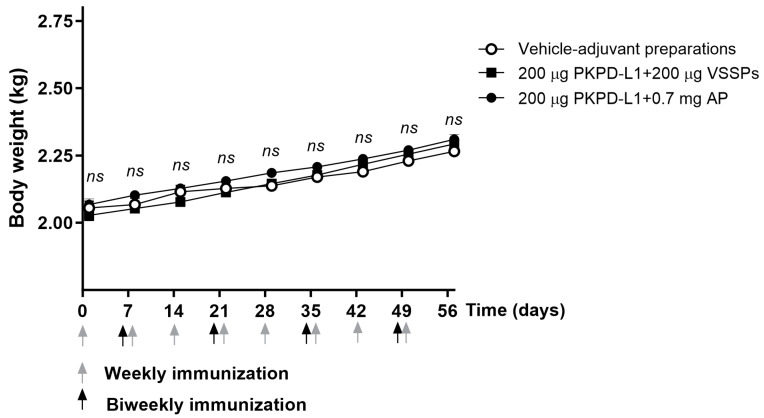
Time course of body weight in rabbits immunized with PKPD-L1. Each symbol represents the average and standard deviation of the body weight of the rabbits (*n* = 4) according to the day of the immunization schedule. Groups: (I) vehicle–adjuvant preparations; (II) 200 μg PKPD-L1 + 200 μg VSSPs; (III) 200 μg PKPD-L1 + 0.7 mg AP. No differences were detected between each immunization group (ns *p* ≥ 0.05, two-way ANOVA, Sidak’s post-test).

**Figure 6 vaccines-13-00296-f006:**
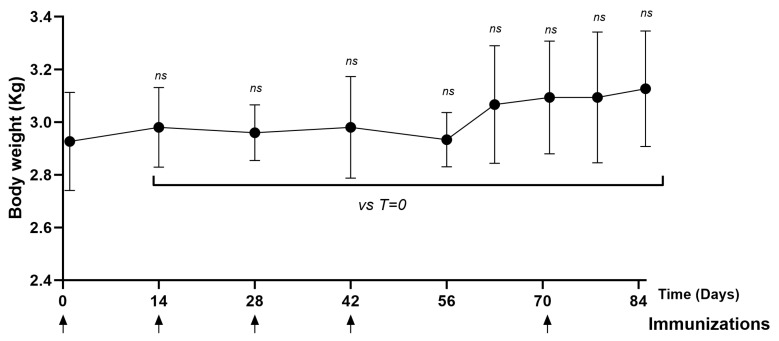
Time course of body weight in NHPs immunized with PKPD-L1. Each symbol represents the average body weight of the NHPs (*n* = 3) according to the day of the immunization schedule. Statistical significance is represented in the graph as ns *p* ≥ 0.05 according to the paired Wilcoxon test (ns: not significant).

**Figure 7 vaccines-13-00296-f007:**
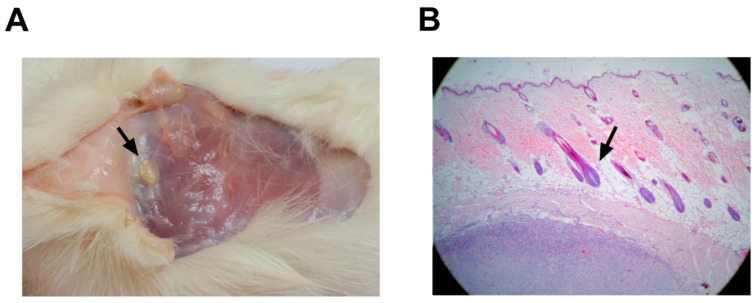
Anatomopathological studies in rats after PKPD-L1^Vac^ immunization. (**A**) Macroscopic findings related to the injection site in rats (small caseous nodules). (**B**) Histological analyses of skin injection site lesions from rats treated with 200 μg of PKPD-L1 in combination with aluminum phosphate (black arrow indicates nodular granulomatous formation). The tissues were stained with hematoxylin and eosin (HE) and observed at 10X.

**Figure 8 vaccines-13-00296-f008:**
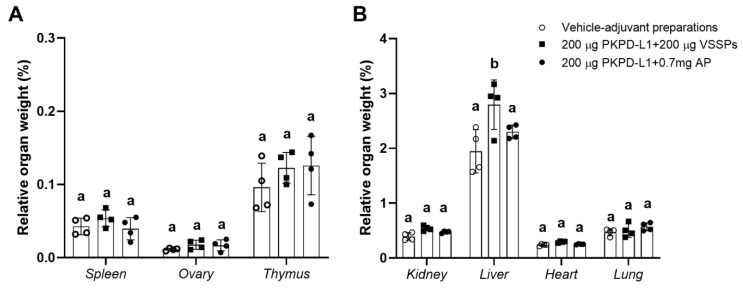
Relative organ weight of selected organs in New Zealand rabbits immunized with PKPD-L1 in combination with VSSPs or aluminum phosphate. Relative weight of small (**A**) and large (**B**) organs. Each symbol represents the weight of the selected organs of the individual rabbits (*n* = 4); data are presented as mean and the standard deviation of the relative organ weight. Different letters indicate statistical significance according to one-way ANOVA and a Tukey’s post-test (*p* < 0.05).

**Table 1 vaccines-13-00296-t001:** Mean and standard deviation of hematological and serum biochemical parameters in rabbits immunized with vaccine PKPD-L1^Vac^.

Parameter	Physiological Range (Unit)	GROUPS	DAY 0 (Pre-Vaccination)	DAY 56 (Post-Vaccination)
White blood cells	5.2–12.5 (10^3^/μL)	I	10.35 ± 2.105 ^a^	15.28 ± 4.484 ^a^
		II	8.4 ± 0.966 ^a^	14.63 ± 2.265 ^a^
		III	11.3 ± 1.386 ^a^	13.33 ± 2.281 ^a^
Red blood cells	5.3–6.8 (10^6^/μL)	I	7.41 ± 0.312 ^a^	6.58 ± 0.5 ^a^
		II	6.64 ± 0.212 ^a^	6.74 ± 0.512 ^a^
		III	6.98 ± 0.137 ^a^	6.96 ± 0.441 ^a^
Hemoglobin	9.8–14 (g/dL)	I	12.58 ± 0.78 ^a^	11.15 ± 0.975 ^a^
		II	11 ± 0.216 ^a^	11.9 ± 1.122 ^a^
		III	11.93 ± 0.499 ^a^	12.05 ± 0.794 ^a^
Hematocrit	34–43 (%)	I	43.9 ± 2.483 ^a^	39.13 ± 3.198 ^a^
		II	39.8 ± 1.087 ^a^	40.78 ± 3.556 ^a^
		III	41.58 ± 1.597 ^a^	41.88 ± 2.947 ^a^
Mean Corpuscular volume	60–69 (fL)	I	59.23 ± 2.141 ^a^	59.45 ± 1.723 ^a^
		II	58.83 ± 0.538 ^a^	60.5 ± 2.04 ^a^
		III	59.58 ± 1.352 ^a^	60.18 ± 1.47 ^a^
Mean Corpuscular Hemoglobin	20–23 (Pg)	I	16.95 ± 0.592 ^a^	16.95 ± 0.436 ^a^
		II	16.58 ± 0.403 ^a^	17.68 ± 0.826 ^a^
		III	17.1 ± 0.469 ^a^	17.35 ± 0.493 ^a^
Mean Corpuscular Hemoglobin Concentration	31–35 (g/dL)	I	28.65 ± 0.289 ^a^	28.45 ± 0.238 ^a^
		II	28.15 ± 0.686 ^a^	29.18 ± 0.457 ^a^
		III	28.68 ± 0.17 ^a^	28.78 ± 0.33 ^a^
Platelet count	158–650 (10^3^/μL)	I	172 ± 78.59 ^a^	515 ± 151.30 ^a^
		II	261 ± 71.58 ^a^	354.25 ± 77.779 ^a^
		III	330.25 ± 87.724 ^a^	286.75 ± 67.465 ^a^
Neutrophils percentage	25–75 (%)	I	58.75 ± 9.535 ^a^	57.25 ± 8.342 ^a^
		II	75.5 ± 7.853 ^a^	55.5 ± 13.304 ^a^
		III	70.5 ± 10.017 ^a^	50.25 ± 9.5 ^a^
Lymphocytes percentage	30–85 (%)	I	38.25 ± 10.996 ^a^	39.5 ± 9.037 ^a^
		II	22.5 ± 7.047 ^a^	41.5 ± 13.304 ^a^
		III	28.75 ± 10.34 ^a^	46.5 ± 9.678 ^a^
Monocytes percentage	1–4 (%)	I	1.25 ± 2.5 ^a^	0.75 ± 0.957 ^a^
		II	1 ± 0.816 ^a^	0.75 ± 0.957 ^a^
		III	0.5 ± 10 ^a^	0.5 ± 0.577 ^a^
Alanine aminotransferase	10–45 (U/L)	I	18.52 ± 6.414 ^a^	27.56 ± 8.241 ^a^
		II	20.13 ± 6.072 ^a^	28.15 ± 5.663 ^a^
		III	18.38 ± 6.363 ^a^	37.63 ± 6.694 ^a^
Aspartate aminotransferase	10–120 (U/L)	I	7.44 ± 2.912 ^a^	22.9 ± 7.506 ^a^
		II	19.69 ± 18.632 ^a^	21.73 ± 1.985 ^a^
		III	12.25 ± 8.012 ^a^	17.94 ± 2.492 ^a^
Creatinine	0.5–2.5 (mg/dL)	I	0.9 ± 0.648 ^a^	2.37 ± 0.287 ^a^
		II	0.63 ± 0.574 ^a^	2.15 ± 0.287 ^a^
		III	0.3 ± 0.600 ^a^	1.52 ± 1.166 ^a^
Total Protein	5.3–8.4 (g/dL)	I	6.94 ± 1.763 ^a^	7.38 ± 0.506 ^a^
		II	6.32 ± 0.517 ^a^	8.05 ± 1.414 ^a^
		III	6.78 ± 0.524 ^a^	8.07 ± 0.760 ^a^
Total bilirubin	1.5–2.5 (mmol/L)	I	1.04 ± 0.39 ^a^	0.76 ± 0.376 ^a^
		II	0.86 ± 0.430 ^a^	0.68 ± 0.364 ^a^
		III	1.2 ± 0.534 ^a^	0.57 ± 0.216 ^a^
Albumin	2.7–5 (g/dL)	I	3.99 ± 0.568 ^a^	4.41 ± 0.812 ^a^
		II	3.2 ± 0.520 ^a^	4.17 ± 0.483 ^a^
		III	2.82 ± 0.314 ^a^	3.78 ± 0.706 ^a^

Note: Values represent the mean ± standard deviation (SD) of the four animals in each group per time-point. Groups: (I) vehicle–adjuvant preparations; (II) 200 μg PKPD-L1 + 200 μg VSSPs; (III) 200 μg PKPD-L1 + 0.7 mg AP. Variables with common letters mean the difference was not statistically significant (ns *p* ≥ 0.05; Wilcoxon test).

**Table 2 vaccines-13-00296-t002:** Mean and standard deviation of hematological and serum biochemical parameters in monkeys immunized with PKPD-L1 antigen in combination with aluminum phosphate.

PARAMETER	Physiological Range (Unit)	DAY 0 (Pre-Vaccination)	DAY 85 (Post-Vaccination)
White blood cells	3.1–9.6 (10^3^/μL)	5.15 ± 0.07 ^a^	6.77 ± 2.19 ^a^
Red blood cells	5.10–7.9 (10^6^/μL)	6.22 ± 0.13 ^a^	6.29 ± 0.14 ^a^
Hemoglobin	11.00–19.04 (g/dL)	12.30 ± 0.44 ^a^	12.30 ± 0.40 ^a^
Hematocrit	33.1–49.67 (%)	42.43 ± 1.27 ^a^	42.83 ± 1.06 ^a^
Mean Corpuscular volume	52.02–76.06 (fL)	68.33 ± 2.52 ^a^	68.33 ± 3.22 ^a^
Mean Corpuscular Hemoglobin	17.1–24.2 (Pg)	19.77 ± 0.80 ^a^	19.57 ± 1.00 ^a^
Mean Corpuscular Hemoglobin Concentration	23.6–33.9 (g/dL)	29.00 ± 0.10 ^a^	28.7 ± 0.17 ^a^
Platelet count	281–672 (10^3^/μL)	375 ± 33.18 ^a^	364.33 ± 9.07 ^a^
Alanine aminotransferase	1–56 (U/L)	11 ± 2.65 ^a^	5.93 ± 3.25 ^a^
Aspartate aminotransferase	10.3–59.8 (U/L)	31 ± 9.17 ^a^	33.27 ± 15.66 ^a^
Creatinine	0.1–1.10 (mg/dL)	0.30 ± 0.05 ^a^	0.34 ± 0.44 ^a^
Alkaline phosphatase	60–850.4 (U/L)	588.33 ± 233.33 ^a^	390 ± 152.59 ^a^
Glucose	56.81–99.09 (mmol/L)	81.08 ± 0.457 ^a^	83.24 ± 0.272 ^a^
Total bilirubin	0.1–2.30 (mg/dL)	3.58 ± 0.414 ^a^	1.70 ± 0.458 ^a^
Direct bilirubin	0.1–0.80 (mg/dL)	0.80 ± 0.173 ^a^	0.77 ± 0.153 ^a^
Triglycerides	9–213 (mg/dL)	175.10 ^a^	178.91 ^a^
Phosphorus	1.10–6.60 (mg/dL)	2.287 ± 0.38 ^a^	1.72 ± 0.13 ^a^
Urea	7.50–28 (mg/dL)	9.05 ± 2.69 ^a^	10.03 ± 0.35 ^a^
Gama-glutamyl transferase	1–199 (U/L)	65.67 ± 24.21 ^a^	76 ± 36.06 ^a^

Note: Values represent the mean ± standard deviation (SD) of the three animals per time-point. Variables with common letter indicates that the differences were not statistically significant (*p* > 0.05; Wilcoxon test).

## Data Availability

The data sets generated during the current study are available from the corresponding author on reasonable request.

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
