# Peer review of "Immunogenicity and Safety Profile of Two Adjuvanted-PD-L1-Based Vaccine Candidates in Mice, Rats, Rabbits, and Cynomolgus Monkeys"

_vaccines, 2025, doi:10.3390/vaccines13030296_

Round 1

Reviewer 1 Report

Comments and Suggestions for Authors

Question 1: What is the explanation for giving the same dose (200ug) of PD-L1 vaccine to different species of different body weights?

Question 2: It is helpful for the reader to note the species in which the experiment was performed.

Author Response

Reviewer 1:  Comments and suggestions for authors:

Query 1:

What is the explanation for giving the same dose (200 µg) of PD-L1 vaccine to different species of different body weights?

Response 1:

The rationale for administering the same dose (200 µg) of the PD-L1 vaccine to different species, despite varying body weights, is grounded in the unique mechanism of action of vaccines compared to other therapeutic agents [1]. Unlike treatments such as chemotherapeutics, monoclonal antibodies, or cytokines, which are distributed systemically and require dose adjustments based on body weight, vaccines primarily rely on the immune system's localized and systemic response to the antigen. This response is determined by the quantity of antigen required to stimulate effective immunity, rather than by the size or weight of the organism. This feature is a distinct advantage of active immunotherapy over traditional cancer treatments.

For many foreign antigens (e.g., viral, bacterial, or synthetic), there is often a linear dose-response relationship. However, this is not consistently observed with therapeutic vaccines, where effective immunogenicity may be achieved with the same dose across species [2].

Examples from the literature support this fact, even evaluating from the preclinical stage the same doses that will be used in humans.

In experimental studies of a vaccine formulation of recombinant human VEGF antigen with aluminum phosphate (call CIGB-247) were evaluated different antigen doses in mice (100 µg, 200 µg, or 400 µg) and monkeys (200 µg or 400 µg) [3]. The results indicated that healthy CIGB-247 vaccinated C57BL/6 mice developed anti-hVEGF IgG antibodies with mean titer values of 1:15 000, 1:29 000, and 1:32 000, for the 100 ug, 200 µg, or 400 µg antigen groups, respectively. In CIGB-247 vaccinated monkeys, anti-hVEGF IgG antibodies reach mean titer values of 1:24 000 and 1:34 000, for the 200 ug, or 400 µg antigen groups respectively [3]. Therefore, similar results were found in the specific response between mice and monkeys regardless of whether the body weight of both species is quite different. In addition, doses of 200 µg and 400 µg of antigen formulated with aluminum phosphate were evaluated in the CENTURO 2 clinical trial in patients with solid tumors [4]. These antigen doses successfully induced humoral immunity and demonstrated safety and immunogenicity in cancer patients. Also, immunogenicity and some safety features of a CIGB-247 in formulation with VSSP were studied in rats, rabbits, and non-human primates using a common 100 µg antigen dose, followed by a dose escalation study in monkeys (100 µg to 200 µg and 400 µg/vaccination) [5, 6]. In phase I clinical trials, vaccination with this formulation (at different doses: 50 µg, 100 µg and 400 µg) induced specific anti-VEGF antibodies in patients with solid tumors [7].

Another example comes from evidence regarding a vaccine candidate that utilizes the extracellular domain (ECD) of HER1. For mice immunization experiments, this vaccine was prepared by mixing EGFR-ECD (50 µg, 100 µg, or 200 µg) with VSSP [8]. In the non-clinical immuno-toxicological evaluation of HER1 cancer vaccine treated monkeys were immunized with 200 µg of the antigen [9]. In phase I clinical trials, vaccination with this formulation (at different doses: 100 µg, 200 µg, 400 µg, 600 µg, and 800 µg) successfully induced humoral immunity and demonstrated its safety and immunogenicity in patients with hormone-refractory prostate cancer. There were no significant differences regarding the geometric means of the anti-HER1 titers among the dose groups except in the group of 100 µg in which antibody titers were significantly lower [10].

For the PKPD-L1 vaccine, to find the optimal dose of the vaccine, a previous study was carried out in mice where a range of doses 50 µg, 100 µg, 200 µg and 400 µg was evaluated (Figure 1). The results indicated that the bests antibody responses were obtained with the higher doses (200 µg and 400 µg). The highest dose evaluated (400 µg) did not significantly increase the specific response to PD-L1 as compare with the 200 µg dose. Then, the 200 µg antigen dose was selected to evaluate the anti-tumor effect in the CT26 and MB16 F10 mouse tumor models, demonstrating PD-L1 vaccine anti-tumor capacity [11]. With this background, in this work, we decided to delve deeper and characterize both the immune response and the possible adverse effects that could appear due to the specific induction of PD-L1.

Figure 1. Antigen dose escalation study of PKPD-L1 vaccine in BALB/c mice. Antibody titers were log-transformed before all statistical processes and are presented as geometric mean titers. Geometric mean titers values of 1:148 000, 1:465 000, 1:2 000 000 and 1:3 800 000 for the 50 µg, 100 µg, 200 µg, or 400 µg antigen groups, respectively. Different letters indicate statistical significance according to one-way ANOVA and Tukey’s post-test.

Based on these results, the 200 µg dose was selected for further investigation into the immune response elicited by the PKPD-L1 vaccine in BABL/c and C57BL/6 mouse strains, as well as in rats, rabbits, and non-human primates. Despite the inclusion of species with varying body weights and the use of a range of doses, the antibody responses (measured as titers) were consistent across the four models. Figure 2, presented below, illustrates the antibody titers observed after last immunization doses using the PKPD-L1 vaccine formulations as per the proposed regimen.

Notably, differences in antibody responses were identified between the two mouse strains when using a specific adjuvant (VSSP). These differences were attributed to immunological and genetic factors inherent to the strains rather than to body weight, which is comparable between them. These findings underscore that the immune response to PKPD-L1 vaccination is primarily influenced by the antigen dose required to elicit effective immunity, rather than by the recipient's body weight. The consistent immunogenicity observed across diverse species validates the use of a standardized dose for preclinical and clinical studies. This approach not only ensures reliable immune activation but also simplifies dosing strategies, emphasizing a significant advantage of therapeutic vaccines in cancer treatment.

Figure 2. Anti-PD-L1 IgG antibody titer in mice BALB/c and C57BL/6, Wistar rats, New Zealand rabbits and Chlorocebus aethiops sabaeus moneys immunized with PKPD-L1 in two different adjuvants. Antibody titers were log-transformed before all statistical processes and are presented as median for 200 µg PKPD-L1 + VSSPs and as geometric mean titers for 200 µg PKPD-L1 + AP. Median titer for 200 µg PKPD-L1 + VSSPs: (BALB/c; 1:300 000), (C57BL/6; 1:50 000), (Wistar rats; 1:150 000), (New Zealand rabbits; 1:135 000). Geometric mean titers for 200 µg PKPD-L1 + AP: (BALB/c; 1:400 000), (C57BL/6; 1:5 000 000), (Wistar rats; 1:300 000), (New Zealand rabbits; 1:230 000), (Moneys; 1:260 000). Different letters indicate statistical significance according to one-way ANOVA follow a Tukey’s post-test for 200 µg PKPD-L1 + AP and Dunn's post-test for 200 µg PKPD-L1 + VSSPs.

Query 2: 

It is helpful for the reader to note the species in which the experiment was performed.

Response 2:

We thank the reviewer for highlighting the importance of clearly specifying the species used in the experiments. We agree that this detail is critical for ensuring transparency and facilitating the interpretation of our findings. Throughout the manuscript, we have made a concerted effort to clearly indicate the species involved, starting from the title, and extending through the abstract, methods, results, and discussion sections.

If there are specific sections where the reviewer feels our descriptions of the species are unclear or incomplete, we would greatly appreciate further guidance. We are fully committed to addressing any gaps and ensuring that the information is as clear and detailed as possible for the reader.

Reviewer 2 Report

Comments and Suggestions for Authors

The manuscript shows some results on the development and evaluation of two PD-L1-based cancer vaccines, examining their immunogenicity (antibody against PD-L1) and preliminary safety across various preclinical models, including BALB/c and C57BL/6 mice, rats, rabbits, and non-human primates. While the subject is timely and aligns with the growing interest in immunotherapy, particularly checkpoint inhibitors like PD-L1, the study raises significant concerns.

Evaluating an anticancer vaccine candidate, especially one incorporating a non-previously evaluated vaccine formulation (using VSSP adjuvant), across multiple species, only showing that it induces anti-PDL1 antibody response and preliminary safety evaluation without presenting data on its anticancer immunogenicity and efficacy challenges the focus and justification of the research.

The primary objective of an anticancer vaccine candidate is to demonstrate its antitumor immunogenicity and efficacy in reducing tumor burden or enhancing survival in relevant cancer models. Omitting these critical results undermines the manuscript's scientific impact, potentially rendering the study incomplete or misaligned in its priorities. Although PDL1 is present in tumor cells and blocking its immunosuppressive effect would be one of the main objectives of the vaccine, it is also present in dendritic cells and at least a result of stimulation of the cellular immune response could have been shown in these preliminary trials. Showing the stimulation of different Ig subclasses against PDL1 and discussing this as evidence of stimulation of Th1/Th2 cellular response sounds too superficial and incomplete in this context.

Moreover, with this design, the application of the 3 Rs principle is questionable by using so many different species for preliminary studies without having shown the desired antitumor pharmacological effect in at least one of them (commonly mice). Moreover, including non-human primates is ethically and scientifically defensible only when robust anticancer efficacy has been demonstrated in lower species, such as mice or rats. Without such basic data, the use of primates may appear premature and ethically contentious. In this regard, the ethical approvals were mentioned, and adherence to the 3Rs principle was stated. However, the rationale for using too many species for a preliminary study (including monkeys) must be justified.

Some sections (e.g., Methods) are overly detailed and might benefit from streamlining, focusing on key aspects relevant to the study outcomes.

Figure 8. The relative weight of organs is not a histopathological evaluation.

With these observations, I consider that this manuscript does not meet the minimum requirements of rigor to be accepted in this journal.

Author Response

Reviewer 2:  Comments and suggestions for authors:

The manuscript shows some results on the development and evaluation of two PD-L1-based cancer vaccines, examining their immunogenicity (antibody against PD-L1) and preliminary safety across various preclinical models, including BALB/c and C57BL/6 mice, rats, rabbits, and non-human primates. While the subject is timely and aligns with the growing interest in immunotherapy, particularly checkpoint inhibitors like PD-L1, the study raises significant concerns.

Query 1:

Evaluating an anticancer vaccine candidate, especially one incorporating a non-previously evaluated vaccine formulation (using VSSP adjuvant), across multiple species, only showing that it induces anti-PDL1 antibody response and preliminary safety evaluation without presenting data on its anticancer immunogenicity and efficacy challenges the focus and justification of the research.

The primary objective of an anticancer vaccine candidate is to demonstrate its antitumor immunogenicity and efficacy in reducing tumor burden or enhancing survival in relevant cancer models. Omitting these critical results undermines the manuscript's scientific impact, potentially rendering the study incomplete or misaligned in its priorities. Although PDL1 is present in tumor cells and blocking its immunosuppressive effect would be one of the main objectives of the vaccine, it is also present in dendritic cells and at least a result of stimulation of the cellular immune response could have been shown in these preliminary trials. Showing the stimulation of different Ig subclasses against PDL1 and discussing this as evidence of stimulation of Th1/Th2 cellular response sounds too superficial and incomplete in this context.

Response 1:

We sincerely appreciate the reviewer’s thoughtful comments, which have prompted us to further clarify critical aspects of our study.

First, this work builds upon prior findings that demonstrated the antitumor efficacy of the PKPD-L1 vaccine candidate. Specifically, we previously reported in Morera-Díaz et al., 2023 ("Active immunization with a structurally aggregated PD-L1 antigen breaks T and B immune tolerance in non-human primates and exhibits in vivo anti-tumoral effects in immunocompetent mouse tumor models") that the vaccine administration induced significant antitumor activity in both CT-26 and B16-F10 primary tumor models in mice. Immunization with PKPD-L1Vac was shown to increase tumor-infiltrating lymphocytes and reduce the proportion of CD3+CD8+PD1+high anergic T cells in CT-26 tumor tissues, suggesting a remodeling effect on the tumor microenvironment [11].

We acknowledge that this critical context was not sufficiently emphasized in the introduction of the current manuscript. Instead, our introduction focused primarily on highlighting the unique contributions and differences presented in this study, particularly with respect to the vaccine’s immunogenicity. To address this, and in line with the reviewer’s valuable feedback, we have revised the introduction to explicitly include this background information (lines 50 to 53). These additions aim to ensure that readers can fully understand the foundation of our work and how it builds upon previous findings.

We value the reviewer’s observations and have incorporated their suggestions to strengthen the manuscript. By revising the introduction and explicitly outlining the rationale for our experimental design, we aim to provide a clearer and more comprehensive context for the study. These adjustments ensure that the antitumor effects previously demonstrated are appropriately highlighted and that our commitment to ethical research practices is transparently conveyed.

Query 2:

Moreover, with this design, the application of the 3Rs principle is questionable by using so many different species for preliminary studies without having shown the desired antitumor pharmacological effect in at least one of them (commonly mice). Moreover, including non-human primates is ethically and scientifically defensible only when robust anticancer efficacy has been demonstrated in lower species, such as mice or rats. Without such basic data, the use of primates may appear premature and ethically contentious. In this regard, the ethical approvals were mentioned, and adherence to the 3Rs principle was stated. However, the rationale for using too many species for a preliminary study (including monkeys) must be justified.

Response 2:

We also wish to clarify the rationale for the selection of species and experimental designs in this study, which are guided by the Reduction Principle of the 3Rs (Replacement, Reduction, Refinement). Our goal is to minimize the number of animals used while obtaining scientifically robust data, adhering to ethical and welfare considerations. By employing multiple species, including mice, rats, rabbits, and monkeys, we aimed to maximize the translational relevance of our findings while minimizing suffering. Each species represents a distinct level of biological complexity, allowing us to evaluate the vaccine’s effects across a diverse set of mammalian models. This approach ensures that the immune and antitumor responses observed are not species-specific but rather generalizable, improving the translational potential of our findings to human applications. Furthermore, we intentionally selected sample sizes that balance statistical validity with the ethical imperative to minimize animal use. Our methodology aligns with the ethical and scientific standards of preclinical research, ensuring that our findings are both rigorous and responsible.

Query 3:

Some sections (e.g., Methods) are overly detailed and might benefit from streamlining, focusing on key aspects relevant to the study outcomes.

Response 3:

We agree with the reviewer on this point. In line with this, M&M sections 2.6.1; 2.6.2; 2.8.1; 2.8.3, and 2.8.4 have been simplified and more detailed information appears in the supplementary files.

Query 4:

Figure 8. The relative weight of organs is not a histopathological evaluation.

Response 4:

We agree with the reviewer. Corrected in the document.

Final reviewer comment:

With these observations, I consider that this manuscript does not meet the minimum requirements of rigor to be accepted in this journal.

Response 5:

We respectfully disagree with the reviewer’s assessment and believe that our manuscript meets the rigor and quality standards required for publication in Vaccines.

Our study presents original research within the rapidly evolving field of immunotherapy and immune checkpoint inhibitors, a topic of significant interest to the readers of Vaccines. Specifically, this is the first report to comprehensively evaluate the potential negative effects of an active immunotherapy targeting PD-L1 across multiple species, utilizing a combination of clinical, biochemical, and histopathological methods. This approach offers a novel perspective on the immunological and safety implications of breaking immune tolerance with a self-antigen-targeted vaccine, a critical consideration for advancing therapeutic vaccines in oncology.

The research was conducted using robust experimental methodologies designed to ensure the reliability and reproducibility of the results. Key aspects include:

  • Multispecies Approach: Our study evaluated the immunogenicity and safety of a PD-L1 vaccine across several species (mice, rats, rabbits, and non-human primates), providing comprehensive data on its cross-species effects and translational relevance.
  • Safety Assessment: We employed clinical, biochemical, and histopathological evaluations to thoroughly assess potential adverse effects, ensuring a rigorous analysis of the vaccine's safety profile.
  • Immunological Insights: By demonstrating how active immunization can effectively overcome immune tolerance to a self-antigen without eliciting concerning safety issues, our research adds valuable knowledge to the field of therapeutic cancer vaccines.
  • Relevance to Therapeutic Vaccines: This study is one of the few to focus on a PD-L1-targeted vaccine, filling a significant gap in the literature and addressing an area of high interest for immunotherapy research.

We believe our findings will resonate with the audience of Vaccines, as they provide new insights into the development and evaluation of therapeutic vaccines targeting immune checkpoints. This work is particularly relevant given the increasing interest in innovative immunotherapies to address cancer and other diseases. We acknowledge the importance of rigorous standards in research publication and are confident that our study adheres to these requirements. The methodologies, results, and conclusions presented in the manuscript were carefully designed and analyzed to ensure scientific validity and reliability. Additionally, we have no preconceptions that raise concerns regarding the safety of this approach, as demonstrated by the extensive safety data included in the manuscript.

The authors appreciate the reviewer’s perspective and the critical evaluation of our work. However, we firmly believe that our manuscript meets the standards for publication in Vaccines and represents a meaningful contribution to the field of therapeutic immunology. Our study not only advances the understanding of active immunotherapy targeting PD-L1 but also provides a strong foundation for future translational and clinical research.

Reviewer 3 Report

Comments and Suggestions for Authors

The article refers to the immunogenicity and safety profile of adjuvanted PD1 vaccine candidates in a preclinical trial involving mice, rats, rabbits, and monkeys. The rationale is adequate for a vaccine trial, and the methods are standard. Some changes and discussions can improve the manuscript, benefiting the reader. I consider that Table 1 and Figures 5B and 6 can be part of a supplemental file since the results can be discussed in the text. Figure 1 is critical. There is a significant difference in the response of C57BL6 mice with the VSSP vaccine, which was not well discussed in the text. Moreover, Figure 1E should be modified so that the values are the same for the four groups, and Figure 1 part B log values should be used for IgG titers. Please also refer to the differences in IgG2b production in BALB/c mice. Figure 2 shows 4 points for Wister rats, not 5. In addition, with the VSSP vaccine, 3 (75 % of your cohort) behave differently, so it is difficult to conceive that there is no difference between the vaccines. Figure 5 is adequate, although please include 56 days in part C. Figure 4 represents only two vaccinations; what happened to the rest? The Tables are acceptable, but please keep the letter for statistical differences. The legend should be informative. In Table 5, the decrease in alkaline phosphatase is important after 85 days; any explanation?

The authors should consider that Kupfer cells are not good antigen-presenting cells probably leukocyte activation in the liver is to other mechanism

Author Response

Reviewer 3:  Comments and suggestions for authors:

The article refers to the immunogenicity and safety profile of adjuvanted PDL1 vaccine candidates in a preclinical trial involving mice, rats, rabbits, and monkeys. The rationale is adequate for a vaccine trial, and the methods are standard. Some changes and discussions can improve the manuscript, benefiting the reader.

Query 1:

I consider that Table 1 and Figures 5B and 6 can be part of a supplemental file since the results can be discussed in the text.

Response 1:

We appreciate the reviewer’s comment and have made the suggested changes. In the revised version of the manuscript, Table 1 and Figures 5B and 6B have been moved to the supplemental materials and are now labeled as Supplementary Table 1, Supplementary Figure 1, and Supplementary Figure 2.

Query 2:

Figure 1 is critical. There is a significant difference in the response of C57BL6 mice with the VSSP vaccine, which was not well discussed in the text.

Response 2:

Also, in agreement with the examiner’s comments we rephrased four paragraphs of Discussion (lines 580 to 598) as follow: A higher response for both parameters were observed exclusively in the C57BL/6 mouse strain when the antigen was combined with aluminum phosphate. Aluminum-based adjuvants are known to preferentially prime Th2-type immune responses, even when recent studies show that depending on the vaccination route they can enhance both Th1 as well as Th2 cellular response [12]. Conversely, while studies with murine models have shown that VSSP is able to induce humoral and cellular responses for different tumor-cell and peptide-antigen vaccines, this adjuvant is more prone to activate antigen specific CTL response.  VSSP induce in vitro IL 12 secretion, DC-maturation and antigen cross-presentation [8, 9]. In the case of BALB/c mice, being a strain classically chosen to evaluate the humoral response to antigens by its polarization towards a Th2 response, it becomes a little more difficult to find differences if these are subtle between two adjuvants that can both induce a humoral response [13, 14]. While C57BL/6 mice had relatively mild systemic humoral immune responses, but cell-mediated immunity activity is relatively higher than in other mice strains. The ratio of B lymphocytes to T lymphocytes circulating in C57BL/6 mice is inversely proportional to other inbred strains and maybe more subtle differences in the immune response can be tuned [15]. These factors may help explain why differences in the antibody response were observed exclusively in this strain, favoring the combination of the antigen with aluminum phosphate.

Query 3:

Moreover, Figure 1E should be modified so that the values are the same for the four groups, and Figure 1 part B log values should be used for IgG titers.

Response 3:

We agree with the examiner comment. Corrected in the document.

Query 4:

Please also refer to the differences in IgG2b production in BALB/c mice.

Response 4:

Also, in agreement with the examiner’s comments we rephrased four paragraphs of Discussion (lines 616 to 626) as follow: The combination of PKPD-L1 with VSSPs adjuvant showed a Th1 pattern in both mouse strains. Particularly interesting are the results in BALB/c mice strain. BALB/c mice is prototypical Th2-type mouse strain. Nevertheless, when the (IgG2a+IgG2b)/IgG1 ratio was determined, as a surrogate of the Th1/Th2 response pattern, mice immunized with the formulations adjuvanted in VSSPs significantly higher than the ones receiving Alum, an indirect indication of TH1 differentiation. These results are consistent with those shown by other authors using VSSPs as adjuvant. VSSPs induce the secretion of inflammatory cytokines, such as IL-12, which induce Th1 polarization of helper T cells. They al-so stimulate the functional activity of specific cytotoxic T cells, a phenomenon that is facilitated by cross-presentation of exogenous antigens and the independence of helper T cells for primary cytotoxic T cell expansion [16].

Also, in the result section were added lines 314 to 319: The (IgG2a+IgG2b)/IgG1 ratio was determined as a surrogate of the Th1/Th2 response pattern. In BALB/c mice immunized with PKPD-L1 combined VSSPs ratio was 1.49 and for PKPD-L1 combined with aluminum phosphate was 0.59. The ratio with PKPD-L1 combined VSSPs was significantly higher than in the aluminum phosphate-adjuvanted formulation (p=0.0043; Unpaired t-test).

Query 5:

Figure 2 shows 4 points for Wister rats, not 5.

Response 5:

We agree with the examiner comment; there was an omission in the text that has been corrected in the new version (lines 167 to 169) as follow: Rat 1.3 serum sample from group immunized with PKPD-L1 combined with VSSPs was rejected as “unsuitable for analysis”, due to insufficient volume for the ELISA procedures.

Query 6:

In addition, with the VSSP vaccine, 3 (75 % of your cohort) behave differently, so it is difficult to conceive that there is no difference between the vaccines.

Response 6:

We acknowledge the examiner’s observation regarding the apparent differences in antibody titers, as most of the rats treated with PKPD-L1/VSSP showed lower anti-PD-L1 titers compared to those receiving PKPD-L1/aluminum phosphate preparations. While this trend was noted, statistical analysis using standard available software did not detect significant differences, likely due to the small sample size and the high variability inherent to outbred populations. In the absence of statistical evidence to support differences in antibody titers, we are unable to draw definitive conclusions on this matter.

Query 7:

Figure 5 is adequate, although please include 56 days in part C.

Response 7:

Figure 5C specifically presents the data for day 56, which corresponds to fourteen days after the fourth immunization. This time point was chosen because it represents the peak of antibody titers following the induction phase. Consequently, it was selected as the optimal time to evaluate the neutralizing activity of the antibodies.

Query 8:

Figure 4 represents only two vaccinations; what happened to the rest?

Response 8:

Following the observation made by the reviewer, we checked the data and whether the axis had been truncated. We corrected it in the document showing the weight values ​​of the animals until day 56.

Query 9:

The Tables are acceptable, but please keep the letter for statistical differences. The legend should be informative.

Response 9:

We keep the letter for statistical differences.

Query 10:

In Table 5, the decrease in alkaline phosphatase is important after 85 days; any explanation?

Response 10:

The reviewer is encouraged to refer to Table 3 in the original document, which has been renumbered as Table 2 in the revised version. This table, titled “Mean and standard deviation of hematological and serum biochemical parameters in monkeys immunized with PKPD-L1 antigen in combination with aluminum phosphate” provides the relevant data. While it is true that alkaline phosphatase values decreased toward the end of the study, it is important to emphasize that these values remained within the physiological range for the species and strain. Furthermore, the absence of any other pathological signs in the animals suggests that this finding cannot be attributed to the administration of PKPD-L1 or any underlying pathology in the animals.

Query 11:

The authors should consider that Kupfer cells are not good antigen-presenting cells probably leukocyte activation in the liver is to other mechanism.

Response 11:

We acknowledge that this critical context was not sufficiently emphasized in the discussion of the current manuscript. We have revised the of discussion to explicitly include this background information (lines 677 to 690). These additions aim to ensure that readers can fully understand the foundation of our work and how it builds upon previous findings. ¨The human liver is an organ with a diverse array of immunologic functions. Hepatocytes although they are not immune cells, express innate immune receptors and can serve as antigen-presenting cells. Also is an important reservoir of macrophages, with 80–90% of the body’s total macrophages consisting of Kupffer cells that reside within the hepatic sinusoids. They can become potent activators of T cells in the presence of other pathogen-associated molecules or inflammatory cytokines. Also, the liver is rich in innate lymphocytes (e.g. NK & NKT cells) are relatively abundant in the liver compared to other tissues of the body [17, 18]. VSSPs are a nanoparticle drug composed of outer membrane vesicles of gram-negative bacteria Neisseria meningitidis and the GM3 ganglioside. VSSP stimulated peritoneal inflammation characterized by increased granulocytes and monocytes, including inflammatory monocytic cells [19, 20]. On the other hand, a therapy aimed at blocking PD-L1 may have a positive effect on the immune system [21]. Therefore, the combination of the antigen with this potent immunomodulatory adjuvant could explain the activation of different immune cells in the liver, thus increasing the weight of the organ¨.

Round 2

Reviewer 2 Report

Comments and Suggestions for Authors

The authors demonstrated significant effort in addressing the deficiencies of the initial version. They clarified relevant points that were previously missing, which initially led to the consideration of using monkeys as unjustified at this stage of the study. Although their use at this stage may still be debatable, considering that, except for this last consideration, the rationale for obtaining data across different species—particularly in primates—provides relevant information for human applications, I accept the proposal and leave the final approval of the manuscript in the editor’s hands.

Reviewer 3 Report

Comments and Suggestions for Authors

The authors have modified the manuscript according to the suggestions. I have no further comments